# Unequal airborne exposure to toxic metals associated with race, ethnicity, and segregation in the USA

John K. Kodros [1] ✉, Michelle L. Bell [2], Francesca Dominici [3], Christian L'Orange[1], Krystal J. Godri Pollitt [4], Scott Weichenthal[5], Xiao Wu [6] & John Volckens [1]

Persons of color have been exposed to a disproportionate burden of air pollution across the United States for decades. Yet, the inequality in exposure to known toxic elements of air pollution is unclear. Here, we find that populations living in racially segregated communities are exposed to a form of fine particulate matter with over three times higher mass proportions of known toxic and carcinogenic metals. While concentrations of total fine particulate matter are two times higher in racially segregated communities, concentrations of metals from anthropogenic sources are nearly ten times higher. Populations living in racially segregated communities have been disproportionately exposed to these environmental stressors throughout the past decade. We find evidence, however, that these disproportionate exposures may be abated though targeted regulatory action. For example, recent regulations on marine fuel oil not only reduced vanadium concentrations in coastal cities, but also sharply lessened differences in vanadium exposure by segregation.

Exposure to outdoor fine particulate matter (particles with aerodynamic diameter less than 2.5 microns; $PM_{2.5}$) is a leading contributor to the global burden of disease[1]; however, exposure to $PM_{2.5}$ is not distributed evenly across racial and ethnic population sub-groups in the United States. A growing body of evidence has found that communities with a high percentage of persons of color and of low socioeconomic status are often disproportionately exposed to higher concentrations of total $PM_{2.5}$[2,3]. This disparity in exposure has been suggested as one of the causes of higher rates of adverse health outcomes, such as cancer and asthma, among these populations[4,5]. However, $PM_{2.5}$ is a complex mixture of chemical components, which vary spatially and temporally. Moreover, several studies suggest that certain chemical components have an increased risk of health outcomes[6,7]. Despite this growing evidence, studies on environmental injustice of exposure to air pollution have, for the most part, relied on total $PM_{2.5}$ mass, and there is relatively less understanding of the

exposure burden among racial groups to the most toxic $PM_{2.5}$ components.

Racial residential segregation (RRS) is the systematic separation of racial or ethnic groups in separate geographical areas[8,9]. RRS has been suggested as an underlying contributor to the disproportionate exposure to environmental stressors and associated increased health risk among the non-Hispanic Black (NHB) population compared to the non-Hispanic White (NHW) population in the US[9,10]. Previous research has documented an increased risk of infant and all-cause mortality[11,12], cardiovascular disease[13,14], COVID-19 mortality rate[15], and pregnancy complications[16] with increasing RRS.

Increased health risks in communities experiencing high RRS may be caused, in part, by disproportionate exposure to environmental stressors, such as $PM_{2.5}$[10]. Existing research on air pollution exposure and health disparities has focused primarily on associations derived from the proportion of minority racial and ethnic individuals present

[1]Department of Mechanical Engineering, Colorado State University, Fort Collins, Colorado, USA. [2]School of the Environment, Yale University, New Haven, CT, USA. [3]Department of Biostatistics, Harvard T.H. Chan School of Public Health, Boston, MA, USA. [4]Department of Environmental Health Sciences, Yale School of Public Health, New Haven, CT, USA. [5]Department of Epidemiology, Biostatistics, and Occupational Health, McGill University, Montreal, Quebec, Canada. [6]Department of Biostatistics, Columbia University, New York, NY, USA. ✉e-mail: jkkodros.research@gmail.com

in a given neighborhood or community[2,3,17–19]; however, the population fraction of minority racial and ethnic groups in a given neighborhood fails to take into account the broader context of the racial distribution in the urban area (for instance, a 30% NHB neighborhood may imply a different neighborhood in a city with 1% NHB and a city with 30% NHB). Several recent studies have begun to associate RRS with total $PM_{2.5}$ mass concentrations and further relate this increased exposure to increased risk of adverse health outcomes[20–25]. Despite the growing evidence of an association of RRS with exposure to particulate air pollution and increased health risk, evidence of the association of RRS with potentially toxic components of $PM_{2.5}$ is limited[20].

Despite existing in only trace quantities, fine particulate metals are known to be toxic chemical components in $PM_{2.5}$[26–28]. While there is still some uncertainty surrounding the physiological mechanisms in which $PM_{2.5}$ affects health, recent evidence has suggested oxidative stress as an important mechanism through which air pollution increases the risk of adverse respiratory and cardiovascular outcomes[29–31]. Metals, in particular, can generate reactive oxygen species, resulting in enhanced oxidative stress[7,32–35]. Exposure to trace metals in ambient $PM_{2.5}$ has been associated with increased rates of cardiovascular and respiratory mortality and hospitalizations[36]. Further, many of the trace metals in $PM_{2.5}$ have known or suspected carcinogenicity (e.g., Pb, Ni, Cr, V, Ti) and/or neurotoxicity (e.g., Al, Pb, Mn, Fe, Cu, V) in humans[27] (https://www.atsdr.cdc.gov/toxprofiledocs/index.html).

While previous research has clearly demonstrated a disproportionate burden of exposure to total $PM_{2.5}$ in communities with a high percentage of persons of color or low socioeconomic status, there is less understanding of potential inequalities in exposure burdens associated with racial residential segregation or known toxic components of $PM_{2.5}$. Given the wide body of knowledge on trace metal toxicity (and developing knowledge on trace metals in $PM_{2.5}$ and adverse health outcome[26,37]), it is essential to understand where and what sub-populations are exposed to these toxic elements. In this study, we examine the association between racial residential segregation with trace metals in $PM_{2.5}$. We focus on racial residential segregation between non-Hispanic Black and non-Hispanic White populations; however, in the Supplemental Material, we investigate this association with racial residential segregation for Hispanic, Asian, and Native American populations relative to non-Hispanic White populations. In the following sections, we present our analysis that estimates: 1) geographic trends in fine particulate metals commonly associated with anthropogenic emission sources (Cu, Zn, Ni, Cr, Pb, and V) and natural emission sources (Fe, Mn, and Ti); 2) differences in concentrations and mass proportions of metals across counties with varying degrees of racial residential segregation; and 3) the associated relative disparities in population-weighted exposure. Subsequently, we discuss the special case of vanadium, which has undergone decreasing temporal trends in concentration across the US due to enhanced shipping regulations[38].

## Results

### Geographic variability in fine particulate metal concentrations and racial residential segregation

We acquired surface monitoring measurements from the Environmental Protection Agency's Chemical Speciation Network (CSN)[39] and Interagency Monitoring of Protected Visual Environments (IMPROVE)[39,40] for several fine particulate metals: Cu, Zn, Ni, Cr, Pb, V, Fe, Mn, and Ti (see Methods for discussion of valid measurements and limit of detection). These metals were chosen due to their known or suspected health effects (e.g., https://www.atsdr.cdc.gov/toxprofiledocs/index.html) and measurements in both monitoring networks. We grouped these metals into two broad categories: the first includes metals largely associated with anthropogenic sources (Cu, Zn, Ni, Cr, Pb, and V) and the second includes metals typically considered

as tracers for natural sources (Fe, Mn, and Ti). This grouping was based on a literature review of source apportionment studies[41–44]. However, we note that these categories are not entirely strict as elements in these groups are not exclusively derived from anthropogenic or natural sources. For instance, Almeida et al.[43] finds that while concentrations of fine particulate Fe in Western Europe are largely emitted through resuspension of mineral dust, there is also an emission source associated with transportation and industrial emissions.

To explore the geographic dependence of particulate metal concentrations across the US, we consider Pb as representative of the particulate metals emitted mostly through anthropogenic sources and Fe as representative of the particulate metals emitted through mostly natural sources. Further, we group states into the geographic regions outlined in Morello-Frosch and Jesdale[20] (Fig. S1). Concentrations of annual mean Pb levels in $PM_{2.5}$ in 2019 ranged from 0.1–5 ng m$^{-3}$ (5th–95th percentiles), while Fe ranged from 10–135 ng m$^{-3}$ (Table S1). Concentrations of fine particulate Pb show a strong geographical distribution, with a statistically significant degree of global spatial autocorrelation (Fig. 1 and Table S1). We find a statistically significant cluster of elevated concentration in the industrial Midwest near the Ohio River Valley, with an average concentration of 3 ng m$^{-3}$, and low concentrations in the Western, Mountain, and Border states, with an average concentration of 1 ng m$^{-3}$ (Fig. 1 and Figs. S2–3).

In contrast, fine particulate Fe concentrations display a lower degree of spatial dependence (Table S1 and Figs. S2–3). Median Fe concentrations in Border states (51 ng m$^{-3}$) are slightly lower than concentrations in the Midwest (78 ng m$^{-3}$). However, after normalizing to $PM_{2.5}$ mass, the mass proportion of Fe is highest in the Border states, likely reflecting the mineral dust source of Fe in the desert Southwest (Fig. 1 and Fig. S2). As a result, concentrations of Fe demonstrate a smaller gradient across urban and non-urban areas compared to Pb (Fig. S1). The ratio of the mean urban-to-nonurban Pb concentration in $PM_{2.5}$ across the US is 4.3 (95th CI: 3.5–5.3) while the respective ratio for Fe is only 2.9 (95th CI: 2.3–3.6). Similarly, when comparing mass proportions of the metals in $PM_{2.5}$, the ratio of urban-to-nonurban Pb mass proportion is 2.1 (95th CI: 1.3-3.0), while the same ratio is 1.5 (95th CI: 1.3–1.7) for Fe mass proportions. The lower dependence on urbanity for concentrations and mass proportions of Fe is likely due to the natural mineral dust emission source, with a similar geographic distribution for concentrations and mass proportions of Mn and Ti. Conversely, Pb is emitted largely through anthropogenic sources in urban areas (similar to Cu, Zn, Ni, Cr, and V).

To examine the association of fine particulate metal concentrations with racial residential segregation, we calculated the dissimilarity index (DI) for all counties with a CSN or IMPROVE monitor (a total of 233 counties). The DI ranges from 0 (indicating perfect evenness) to 1 (indicating complete separation of NHB from NHW). We calculated the DI based on the proportion of NHB and NHW populations at the census tract level relative to the county level (Fig. S2; see Methods). In the Supplemental Material, we also include the DI calculated for Hispanic, Asian, and Native American populations relative to the NHW population (Fig. S2).

We estimated the strength of the association between DI and fine particulate metal concentrations using a univariate linear regression model. Here, we express the slope of the regression as the percent change in the metal concentration associated with a 10% increase in DI (Table S2). Concentrations of Pb increase by 9% (95th CI: 5–13%) per 10% increase in DI, a slightly larger increase than Fe (7%, 95th CI: 4–9%) or total $PM_{2.5}$ (5%, 95th CI: 3–7%). Across the components considered here, the particulate metals commonly associated with anthropogenic emissions are associated with a larger increase in concentration (9–16%) per a 10% increase in DI than the particulate metals associated with natural emissions (4–7% increase in concentration per 10% increase in DI, Table S2). We find a similar relationship after controlling for the geographic regions discussed above (Table S3). Moreover, this

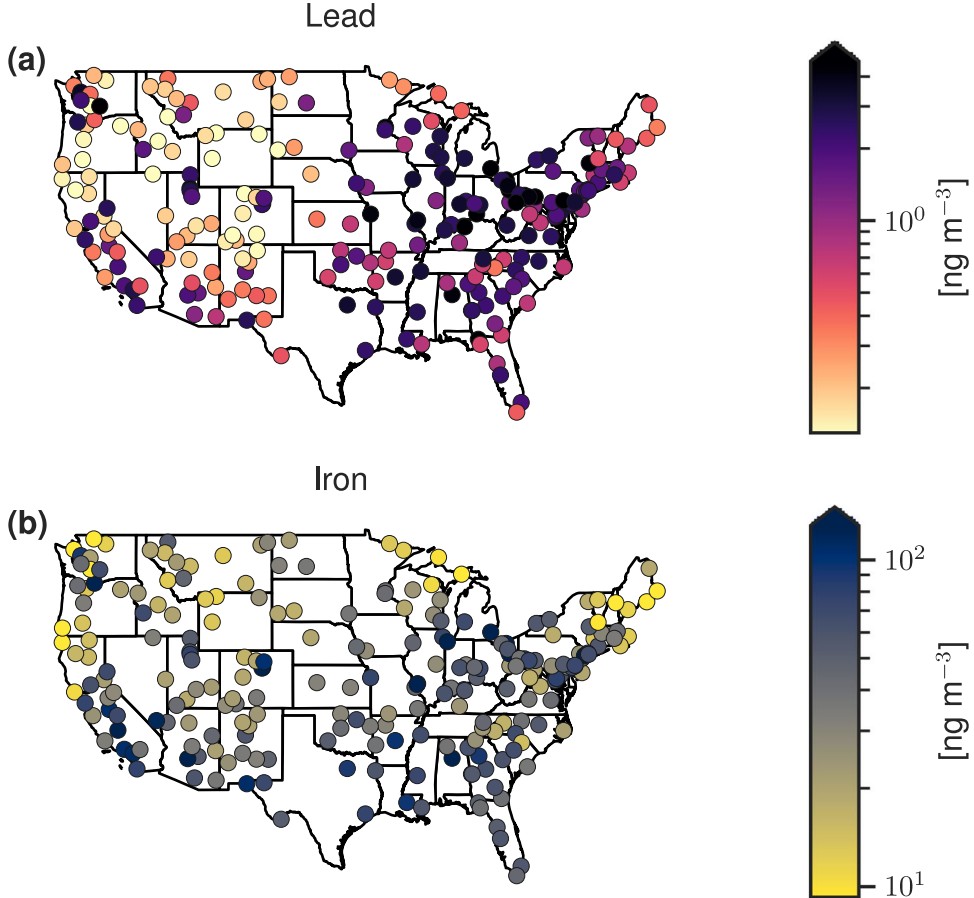

**Fig. 1 | Geographic distribution of fine particulate metals. a** Lead (representing anthropogenically-emitted metals) and **b** iron (representing non-anthropogenically-emitted metals) annual (year 2019) mean mass concentrations in PM$_{2.5}$.

statistical relationship holds for the metals emitted through anthropogenic processes when considering only urban sites, which suggests that our findings are not only the result of urban/rural differences in patterns of RRS. A 10% increase in DI for urban counties is associated with a 5% (95th CI: 1-9%) increase in Pb concentrations, an 11% (95th CI: 6–17%) increase in Zn concentrations, and a 10% (95th CI: 4–15%) increase in Cr concentrations. Conversely, the metals primarily derived from natural emission sources do not all show an increasing association with DI when considering only urban measurement sites. For instance, a 10% increase in DI is associated with an 8% (95th CI: 3–13%) increase in Fe concentrations, but a 0.1% decrease (95th CI: −6–4%) in Ti. We find similar positive associations of DI with fine particulate metal concentrations for the Hispanic, Asian, and Native American populations relative the NHW population (Table S3).

In addition to this association of the DI with fine particulate metals, we also find an association between race/ethnicity and fine particulate metal concentrations. In counties where the percent of the population that identifies as NHB is greater than the national average across all counties, the concentrations of fine particulate metals are consistently elevated relative to counties with a higher than average NHW or Native American population (Fig. S6). This same association is also seen for PM$_{2.5}$, in good agreement with previous studies[45]. Further, we examine the interaction between RRS and racial/ethnic group makeup in each county by expanding the linear model to include an additional variable for the percent of the population identifying as NHB (and a separate model for NHW). We find that concentrations of fine particulate metals commonly associated with anthropogenic emissions increase by 4–8% per 10% increase in DI and 4–6% per 10% increase in NHB population (Table S5). Conversely, these same metals

increase by 9–15% per 10% increase in DI yet decrease by 4–10% per 10% increase in NHW population (though the latter coefficient is not statistically significant for all metals; Table S6). Thus, counties with a high degree of RRS and high NHB population tend to be exposed to higher concentrations of fine particulate metals than counties with a high degree of RRS and a high NHW population.

### Disproportionate burden of metals in highly segregated counties

We categorized counties into three RRS levels as discussed in previous studies[46–48]: well integrated (a DI between 0–0.3), moderately segregated (0.3–0.6), and highly segregated (0.6–1). In this context, a moderately segregated county is one in which 30–60% of either population group would need to relocate to achieve a population distribution across census tracks that matches the county as a whole. Of the counties with CSN/IMPROVE monitors 7% are in well-integrated counties, 67% are in moderately segregated counties, and 19% are in highly segregated counties for the period 2014–2018. The remaining counties contain only one census tract and are excluded from this calculation.

Fine particulate metal concentrations show a strong association with these RRS categories. Concentrations of metals tend to be elevated in highly segregated counties relative to well-integrated and moderately segregated counties (Fig. 2, Fig. S3). This association is strongest across the metals commonly associated with anthropogenic emission sources compared to total PM$_{2.5}$ and metals commonly associated with natural emission sources (Fig. S3). Mean concentrations of Pb in highly segregated counties are a factor of 5 (95th CI: 3-8) higher than in well-integrated counties and a factor of 1.3 (95th CI:

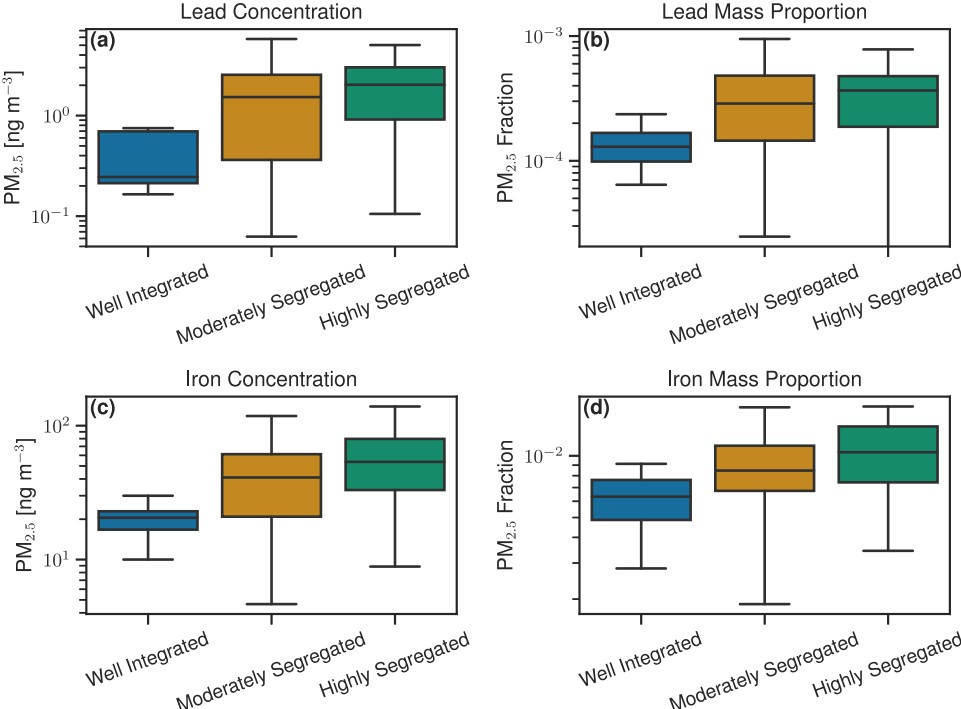

**Fig. 2 | Exposure distributions for fine particulate metals across RRS categories.** The top row depicts lead PM$_{2.5}$ **a** concentrations and **b** mass proportions (representing anthropogenically dominated-emitted particulate metals). The bottom row depicts iron PM$_{2.5}$ (**c**) concentration and **d** mass proportions (representing naturally dominated-emitted particulate metals). RRS categories include: well-integrated ($n = 16$), moderately segregated ($n = 165$), and highly segregated ($n = 47$) US counties for the period 2014–2018. The box represents the interquartile range, the centerline shows the median, and the whiskers are 1.5 times the interquartile range. Outliers have been omitted for clarity.

1.0–1.7) higher than in moderately segregated counties. The mean concentration of Fe is a factor of 3 (95th CI: 2.4–4.3) higher in highly segregated counties compared to well-integrated counties and 1.4 (95th CI: 1.0–1.8) higher than moderately segregated counties (Figs. S5-6). Across all metals associated with emissions from anthropogenic sources the ratio of the mean concentration in highly segregated counties to well-integrated counties ranges from 4 to 20, while the same ratio for metals associated with natural emission sources is less, ranging from 2 to 3 (Fig. S5). Similarly, concentrations of total PM$_{2.5}$ also tend to increase in highly segregated counties by a factor of 2.0 (95th CI: 1.6–2.5) relative to well-integrated counties and a factor of 1.2 (95th CI: 1.0–1.3) higher than in moderately segregated counties (Fig. S7).

The ratios of mean concentrations of fine particulate metals are also higher in highly segregated counties compared to well-integrated counties across RRS categories for Hispanic (ranging from a factor of 2–6), Asian (1–4), and Native American (2–7) populations relative the non-Hispanic White population (Fig. S5); however, comparing the ratio of mean concentrations between highly segregated and moderately segregated counties, only the Native American population (1–3) shows consistently elevated concentrations (Fig. S6).

Mass proportions of trace metals emitted from anthropogenic sources also show a strong association with RRS categories (Fig. 2, Fig. S4). Mean PM$_{2.5}$ mass proportions of Pb in highly segregated counties are a factor of 3 (95th CI: 2–6) higher than in well-integrated counties and are similar to average mass proportions in moderately segregated counties (95th CI: 0.9–2.5). Across all fine particulate metals associated with anthropogenic emissions, the average mass proportion in highly segregated counties is 3-12 times higher than in well-integrated counties and 1.5–1.8 times higher than in moderately segregated counties, though in the latter the 95th CI for lead is below 1. Conversely, mass proportions of the naturally emitted metals tend to be similar across RRS categories (Fig. S4). Iron mass proportions are a

factor of 2 (95th CI: 1–2) higher in highly segregated counties relative to well-integrated counties. For Mn and Ti, this association is less strong. The mean PM$_{2.5}$ mass proportion for Ti is a factor of 0.9 (95th CI: 0.7–1.2) lower in highly segregated counties relative to well-integrated counties. This underscores the relationship of emission source to concentration and PM$_{2.5}$ content across RRS categories. Fine particulate metals dominated by anthropogenic emission sources have a steeper gradient in concentration and mass proportions across RRS categories than metals with a strong natural emission source contribution. Total PM$_{2.5}$, which has both natural and anthropogenic source contributions, is in the middle.

The discrepancy between the metals derived from anthropogenic emissions and natural emissions is likely due to the geographic distribution of the emission sources. The metals with a strong contribution of a mineral dust source tend to have elevated concentrations and mass proportions in rural areas in the Western US, specifically in the Southwest. Conversely, anthropogenic emission sources of metals tend to come from industrial and metallurgical processes or vehicle engines and tire wear. These emission sources tend to be more localized relative to natural emission sources (such as wind-blown soil) and concentrated in urban areas, specifically in inner cities which have long had a history of RRS. Concentrations of Pb are highest around the Ohio River Valley (Fig. 1), which coincides with the highest DIs in the country.

## Relative disparities in exposure to toxic particulate metals

We compared the relationship between racial residential segregation and fine particulate metal exposure by calculating the relative disparity, defined as the coefficient of variation across the population-weighted mean concentrations or mass proportions in highly segregated, moderately segregated, and well-integrated counties, for each chemical component[45] (see Methods). This calculation quantifies differences in exposure across populations living in the different degrees of segregation relative to the average across the entire population,

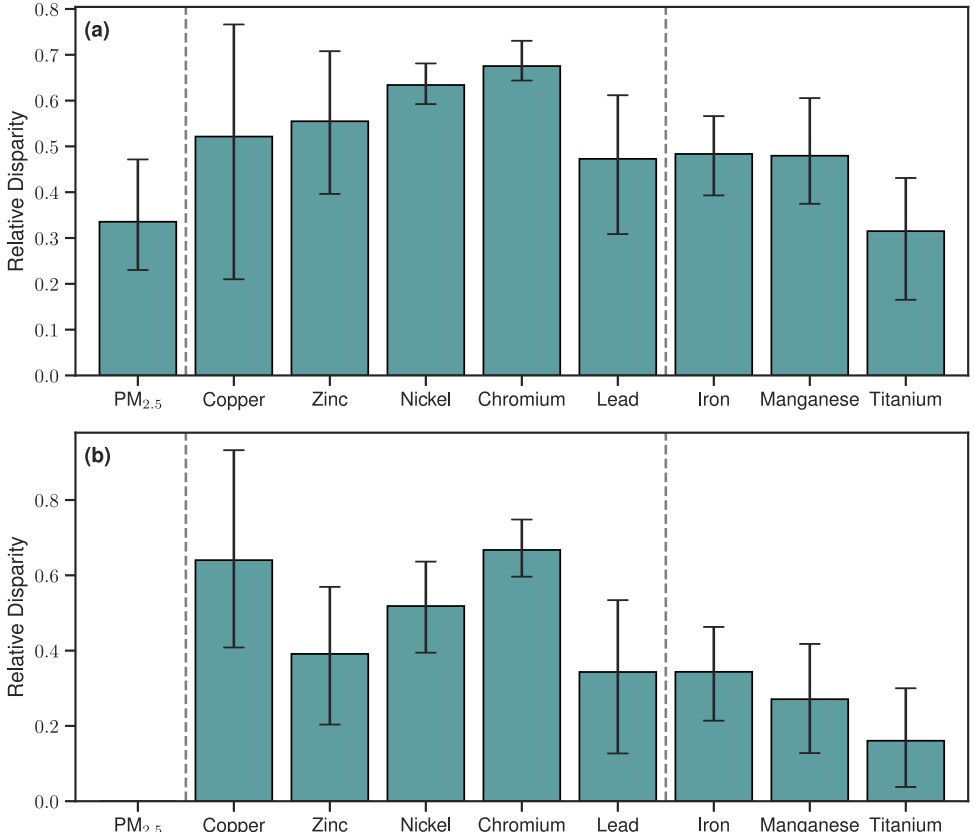

**Fig. 3 | Relative disparities across highly segregated, moderately segregated, and well-integrated US counties.** Relative disparities calculated from **a** population-weighted annual mean concentrations and **b** mass proportions in the year 2019. The relative disparity quantifies the variation in the population-weighted exposures (concentrations or mass proportions) for populations living in each RRS category relative to the average across the entire population (see Methods). This relative scale allows for a direct comparison between the different chemical components considered here. A value of 0 indicates perfect equality in population-weighted exposure across levels of segregation, while increasing values indicate increasing degrees of disparity. Data is presented with the height of the bars representing the estimate of the mean and the error bars representing the 95th percentile confidence interval. Confidence intervals were estimated using 10,000 random samples with replacement (bootstrap sampling) of size $n = 47$, 165, and 16 representing highly segregated, moderately segregated, and well-integrated counties, respectively.

thus allowing for a direct comparison of disparity between different chemical components and time periods independent of magnitude.

The relative disparity across RRS populations exposed to Pb is 0.47 (95th CI: 0.30–0.6), higher than the relative disparity in population-weighted mean exposure to total $PM_{2.5}$, with a relative disparity of 0.33 (95th CI: 0.32–0.47; Fig. 3). While there is an association between RRS and total $PM_{2.5}$ (with concentrations tending to be elevated in highly segregated counties by a factor of 2 over well-integrated counties; Fig. S7), the disparity in concentration across RRS categories is greater for metals commonly associated with anthropogenic emissions. This is likely due to the spatial distribution of emission sources for these metals, as such sources are often found in urban areas with high degrees of RRS. Conversely, total $PM_{2.5}$ mass represents a mixture of sources, not all of which are related to anthropogenic emissions. Therefore, populations living in areas of high RRS are exposed to higher concentrations of total $PM_{2.5}$ mass, as well as higher concentrations of the toxic components of $PM_{2.5}$ considered here compared to people living in areas of low RRS.

The average relative disparity in trace metal $PM_{2.5}$ mass proportions are higher for the metals associated with anthropogenic emission sources (0.3–0.7) compared to those emitted from natural sources (0.2–0.3), though the 95th CI overlap (Fig. 3). Mass proportions of metals commonly associated with natural sources are elevated in non-urban and less-segregated counties due to the large contribution of the mineral dust source to atmospheric load. While total concentrations of the naturally-emitted fine particulate metals considered by this

study tend to be higher in highly segregated counties, normalizing by $PM_{2.5}$ mass generally reduces this association. In contrast, the relative disparities in population-weighted mean $PM_{2.5}$ mass proportions for metals commonly associated with anthropogenic emission sources remain elevated. Thus, even on a per-mass basis, populations living in communities with high RRS are exposed to a form of $PM_{2.5}$ with a higher content of these toxic metals. Moreover, this pattern is roughly the same in previous years (Fig. S8), showing that the relative disparities for most metals have not changed greatly in the past decade.

## Targeted emission reductions can improve segregation-focused disparities in metal exposure

The North American Emissions Control Area and the California Air Resources Board enacted regulations to limit the sulfur content of marine fuel oil used in shipping over a period between 2010 through 2015[38,49]. As a byproduct of these enhanced regulations, the marine fuel oil used to meet the new sulfur regulations also contained a lower proportion of V[38]. Significant reductions in V concentrations related to fuel oil combustion have been observed in the San Francisco Bay Area and near coastal monitoring sites across the US following the enactment of these regulations[44,50–52]. Spada et al.[38] note sharp decreases in V concentrations measured at IMPROVE monitoring sites, especially in coastal cities, directly following the regulations.

Here, we expand upon this analysis considering population-weighted concentrations of V across RRS categories. In 2010, population-weighted mean concentrations of V in highly segregated

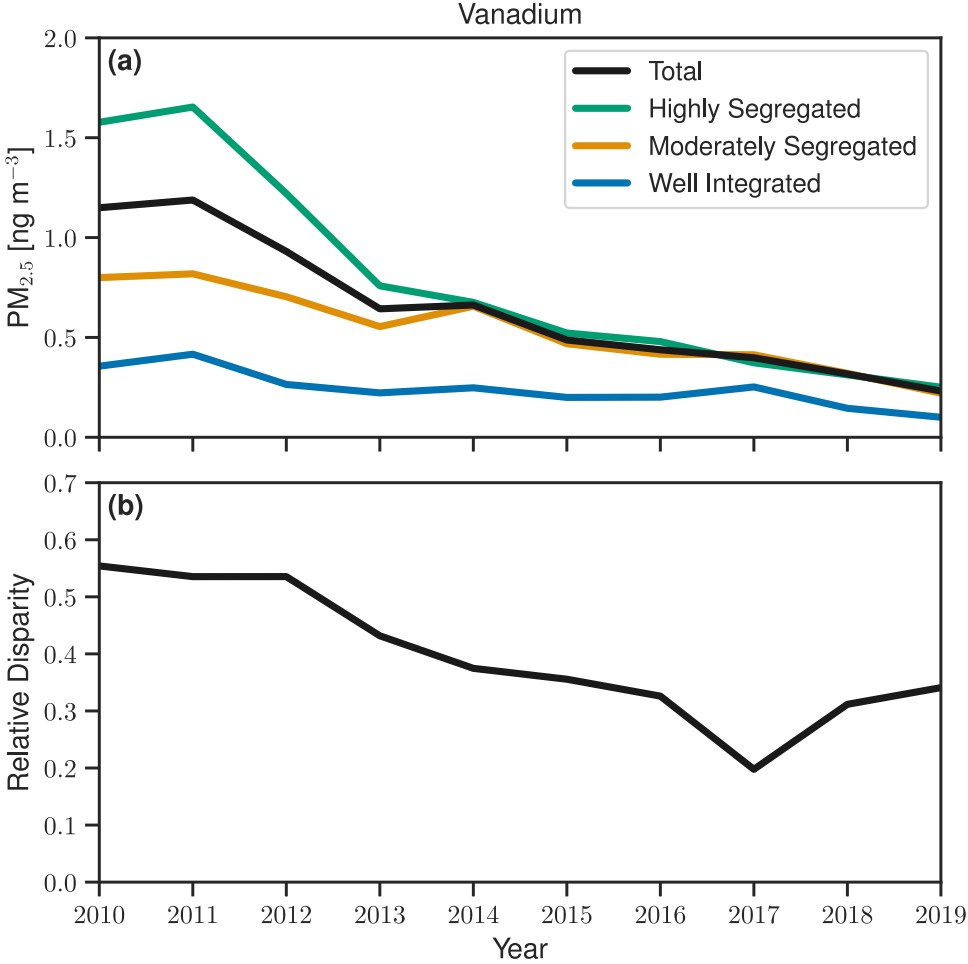

**Fig. 4 | Decreasing temporal trends in Vanadium disparity. a** Population-weighted annual mean V concentrations and **b** relative V disparity across RRS categories.

counties were a factor of 5 (95th CI: 2–9) higher than in well-integrated counties and a factor of 2 (95th CI: 1–3) higher than in moderately segregated counties (Fig. 4). Over the time period between 2010 and 2019, total population-weighted mean concentrations of V across the US steadily decreased as a result of the enhanced shipping regulations, in good agreement with Spada et al.[38] and Hennigan et al.[52]. Notably, the largest contribution to this decrease occurs in the population-weighted mean concentrations in highly segregated counties. By 2019, the population-weighted mean concentration of V in highly segregated counties is roughly equal to that in moderately segregated counties, and is only a factor of 3 (95th CI: 0.9–9) higher than in well-integrated counties.

This decreasing trend is mirrored in the relative disparity in population-weighted mean concentrations of V across RRS categories (Fig. 4). In 2010, the relative disparity in population-weighted mean V concentrations was 0.60 (95th CI: 0.3–0.8), roughly in line with the other metals associated with anthropogenic emissions. By 2019, the relative disparity had decreased to 0.4 (95th CI: 0.1–0.6). This demonstrates that decreasing anthropogenic emissions of particulate metals might reduce the relative disparity in exposure across populations living in segregated and nonsegregated counties, depending on the distribution of those emissions reductions.

## Discussion
While several studies have established a disproportionate exposure to PM$_{2.5}$ mass across racial and ethnic lines, much less is known about exposure burdens associated with racial residential segregation or to the specific toxic components of PM$_{2.5}$. In this study, we find that not

only are counties with higher degrees of racial residential segregation exposed to higher concentrations of total PM$_{2.5}$, they are also exposed to higher mass concentrations of fine particulate metals. Moreover, the relative disparity in exposure to particulate metal concentrations is higher than that for total PM$_{2.5}$. In short, different populations experience "different kinds of particles". Our analysis suggests that a 10% increase in the dissimilarity index for the non-Hispanic Black and non-Hispanic White populations is associated with a 9–16% increase in annual mean concentrations of fine particulate metals associated with anthropogenic emissions, compared to a 5% increase in total PM$_{2.5}$ concentrations. These anthropogenically emitted particulate metal concentrations are on average 30–75% higher in highly segregated counties compared to moderately segregated counties and a factor of 5-20 times higher in highly segregated counties compared to well-integrated counties. Further, PM$_{2.5}$ mass proportions of trace metals are on average 3–12 times higher in highly segregated counties than in well-integrated counties and 1.4–1.8 times higher than in moderately segregated counties, though the 95th CI of the latter for individual metals is below 1. Many of these metals are known carcinogens and/or redox active, which suggests that populations living in areas of high degrees of racial residential segregation are exposed to a higher burden of this type of environmental stressor, which previous studies have suggested may increase the risk of adverse health outcomes. This in turn may contribute to the higher risk or morbidity and mortality in segregated communities with high non-Hispanic Black populations[26,27,29,31,53].

Comparing disparities in fine particulate metals commonly associated with anthropogenic emissions with those emitted through

natural sources suggests that the association between racial residential segregation and concentrations (and mass proportions) is dependent on emission source categories. Source apportionment studies have suggested that fine particulate Cu, Zn, Ni, Cr, Pb, and V are largely, though not necessarily entirely, emitted through anthropogenic sources such as industrial and metallurgical emissions, vehicle engines and tire wear abrasion, or heavy fuel oil combustion in shipping emissions[41,43,44,54]. Conversely, fine particulate Fe, Mn, and Ti are largely emitted through natural processes such as the re-suspension of crustal material. While these classifications are not strict (for instance Fe may also be emitted through metallurgical processes in urban environments)[43], they serve as useful categories in understanding the geographic distribution in concentration. Concentrations and mass proportions of fine particulate metals associated with anthropogenic emissions have a stronger association with racial residential segregation than metals associated with natural sources as well as a stronger association than total $PM_{2.5}$. Anthropogenic emission sources of these metals tend to be concentrated in urban centers, which have historically been associated with high rates of racial residential segregation. In contrast, total $PM_{2.5}$ is a mixture of anthropogenic and natural emission sources. This may in part explain the lower association of total $PM_{2.5}$ with racial residential segregation than for metals associated with anthropogenic emissions.

There are a number of limitations in this study that we address through sensitivity analyses. First, the higher emissions and concentrations of particulate metals associated with anthropogenic sources in urban areas compared to rural areas may bias our results, as urban counties are also often associated with the highest degrees of segregation in our dataset. To account for this, we estimated the strength of the association of RRS with metal concentrations in the subset of data containing only urban measurement sites. When considering only urban sites, we still find an increasing trend in fine particulate metal concentration (a 5–11% increase) for a 10% increase in dissimilarity index for the metals associated with anthropogenic sources, suggesting these results are not solely due to urban/rural gradients in segregation (Table S2). Second, we note that a limitation of this study is the use of discrete monitoring stations with limited spatial coverage across the US. Importantly, the spatial coverage often does not include more than one monitoring site within a county making quantifying within-county variability challenging. This network of sites under-represents rural and low-DI counties. The counties included in our analysis have an average dissimilarity index 0.075 higher (95th CI: 0.05–0.09) than all counties in the US. This monitoring network particularly under-represents counties classified as well integrated (a DI less than 0.3) for the non-Hispanic Black and non-Hispanic White populations. Characterizing counties with a DI less than 0.3 as "well integrated" is based on previous studies; however, a relatively small number of counties characterized as "well integrated" have a CSN/IMPROVE monitor. To test the sensitivity of our results to RRS groupings, we re-define the RRS levels based on three equal percentile groupings of DI across all US counties and three equal percentile groupings of DI across counties with a CSN/IMPROVE monitor. We find that across these different definitions the average concentration of fine particulate metals (especially those associated with anthropogenic emissions) are elevated in the highest DI tier compared to the lowest (Fig. S9). Expanding the CSN/IMPORVE network to allow for within-county variability along with additional rural and low-DI counties would strengthen subsequent studies. Third, the annual mean concentration of a number of the trace metals are often at or near the reported minimum detectable limit (MDL) for the station. As a sensitivity test, we repeated our analysis replacing all values below the MDL with the MDL divided by the square root of two and found this imputation strategy does not change our overall conclusions (Fig. S10). Fourth, while in this study we focus on racial residential segregation between the non-Hispanic Black and non-Hispanic White populations,

we note similar, though less strong, positive associations with trace metal concentrations appear across degrees of racial residential segregation for the Hispanic, Asian, and Native American populations relative the non-Hispanic White population (Fig. S5-S6). In addition, we tested an integrated dissimilarity index intended to capture the degree of RRS across all racial and ethnic groups[20]; this approach yielded similar conclusions. Finally, in calculating DI and degrees of RRS, we rely on arbitrary geographies, census tracts and counties, which may not accurately reflect the spatial distributions of neighborhoods. The DI has noted limitations as an aspatial index of segregation in that the index for a given county is invariant to the spatial distribution of the census tracks and values of the DI may be different with a different choice of spatial scale[55]. Further, this metric relies solely on place of residence and does not take into account the time people may spend in different neighborhoods (such as for work).

While total $PM_{2.5}$ mass concentrations generally shows a decreasing temporal trend in the US with the exception of wildfire-impacted areas, long-term trends in fine particulate metals vary by component and location[38,44,52]. Our results find that the disparity in exposure to particulate metals across RRS categories has persisted over the past decade for all metals considered here except for V. Enhanced regulations targeting the sulfur content of marine fuel oil (which as a by-product also reduced the V content in fuel), resulted in drastic reductions of V concentration in highly segregated counties. By 2019, the relative disparity in the population-weighted mean V concentration across RRS categories was reduced by a factor of 1.5 relative to 2010 (the start of regulations). While these regulations did not intend to specifically target V emissions, it nevertheless demonstrates that targeted emission reductions of anthropogenic sources of metals may have a substantial potential to reduce differences in exposure to these highly toxic components in $PM_{2.5}$ across counties with varying degrees of RRS, depending on the distribution of those emissions reductions.

## Methods
### Air pollution monitoring data
We downloaded daily measurements for concentrations of fine particulate metals (Cu, Zn, Ni, Cr, Pb, V, Fe, Mn, Ti) and total $PM_{2.5}$ for all CSN and IMPROVE monitors for the years 2010 through 2019. Measurements of the metals reflect their concentrations within $PM_{2.5}$ (as opposed to concentrations as a separate particle). Data were obtained from the Federal Land Manager Environmental Database (http://views. cira.colostate.edu/fed/). We kept all measurements labeled as a valid sample (i.e., no null codes or missing values). Measurements for $PM_{2.5}$ and trace metals are reported every third day, leading to an expected number of annual measurements of approximately 121 samples. In averaging the daily samples into an annual average, we kept only the monitors reporting greater than 50% of the expected measurements. We kept all valid measurements below the minimum detectable limit so as not to skew the data towards higher concentrations (following the discussion in Spada et al.[38]). However, we do note that a high percentage of samples for several metals are below the respective minimum detectable limit for that station (Table S1). As a sensitivity test, we repeated our analysis replacing all values below the limit of detection with the limit of detection divided by the square root of 2. While this imputation strategy does lead to minor changes in the quantitative results it does not change our overall conclusions (Fig. S10). Monitoring sites were classified as urban or rural based on urban area classifications from the US Census Bureau. In counties with more than one CSN or IMPROVE monitor, concentrations of the respective pollutants were averaged across the monitors. In 2019, only 21 counties had more than one CSN or IMPROVE monitor (which met our inclusion criteria) and of these only six had more than two monitors. The relative standard deviation in concentration in fine particulate metals and total $PM_{2.5}$ mass ranged from 0.3-140% across these

counties with more than one monitor indicating that there is likely important variability in fine particulate metal concentrations within each county.

## Dissimilarity index

The dissimilarity index (DI) is a measure of the evenness component in RRS when comparing a smaller geographical area to a larger one. The DI ranges from 0 to 1, and represents the fraction of the minority population that would need to move to achieve complete evenness in RRS across the larger geography (Fig. S2). We calculated the DI as:

$$D = 0.5 \times \sum_{i=1}^{n} |\frac{x_i}{X} - \frac{y_i}{Y}| \tag{1}$$

where $x_i$ is the minority population in the smaller geographical unit, $X$ is the minority population in the larger geographical unit, $y_i$ is the reference population in the smaller geographical unit, and $Y$ is the reference population in the larger geographical unit[8,47].

To calculate the dissimilarity index (DI), we downloaded American Community Survey (ACS) summary tables for racial/ethnic population characteristics from the US Census Bureau (https://data.census.gov/cedsci/). We calculated DI based on census tracts as the smaller geographic unit and counties as the larger geographic unit. If a county has only one census tract, we exclude that county from the analysis. We chose to represent RRS at the county level to reduce numerical noise due to low population sizes at smaller geographical units. Census tract level population demographics are only available from the 5-year ACS data. We calculate DI for two 5-year time periods: January 2014 through December 2018 and January 2010 through December 2013 (Fig. S2). While the use of these broad time horizons is a limitation in this analysis, we note that the DI does not change substantially between the two time periods (the average DI across the US over each period is the same). In the main text, we focus our analysis on the DI with non-Hispanic Black (NHB) as the minority population and non-Hispanic White (NHW) as the reference population; however, in the Supplemental Material we present additional analysis with the Hispanic, Asian, and Native American populations as the minority population and NHW as the reference population.

## Calculation of population-weighted means and relative disparities

We calculated population-weighted mean concentration and mass proportions for each RRS category for all particulate metals and for total $PM_{2.5}$ concentration using the following equation:

$$Y_i = \frac{\sum_{j=1}^{J} P_j x_j}{\sum_{j=1}^{J} P_j} \tag{2}$$

where $Y_i$ is the population-weighted average concentration (or mass proportion) for all counties with a CSN or IMPROVE monitor in RRS category $i$, $P_j$ is the population in county $j$, and $x_j$ is the annual mean concentration (or mass proportion) of a given $PM_{2.5}$ component measured in county $j$. If more than one monitor exists in a given county, we averaged them together.

We note two limitations with our calculation of population-weighted mean concentrations and mass proportions. First, this calculation represents only the population living in a county with a CSN or IMPROVE monitor as opposed to the entire US population. Second, we assume the measurement from the surface monitor in a given county is representative of concentrations of the entire county. The spatial heterogeneity of pollution levels could further differ by source and chemical structure of particles.

Relative disparities in population-weighted mean concentrations across RRS categories are calculated through the coefficient of variation following the discussion in Jbaily et al.[45]:

$$RD = \frac{\sqrt{Var(Y)}}{\mu(Y)} \tag{3}$$

where $Y$ is a vector containing the population-weighted mean concentration (or mass proportion) for each RRS category (i.e., $Y_1$, $Y_2$, $Y_3$ from Eq. 2), $Var(Y)$ is the variance in $Y$, and $\mu(Y)$ is the mean of $Y$. The strength of the coefficient of variation is that it measures the spread of the data independent of the data magnitude, thus allowing for direct comparisons of population-weighted concentrations and mass proportions between different chemical components and years.

## Statistical analysis

We used a univariate linear regression model to test the strength of the association between DI and particulate metal (and total $PM_{2.5}$ mass) concentrations (where both DI and pollutant concentrations are log-transformed). Results of the model are expressed as the predicted percent change in concentration associated with a 10% increase in DI (Table S2). We tested this association for all monitor sites (that meet selection criteria) and for sites in urban areas. We assumed statistical significance for coefficients at the 95th percentile confidence interval (i.e., a $p$ value less than 0.05). We expanded upon this univariate model in a number of ways. First, we stratified the model by urban/rural classification from the US Census Bureau. Next, we included a categorical fixed effects variable for geographic regional grouping of states. Finally, we added the percent NHB population proportion as an additional variable (as well as NHW in a separate model). All data processing and analyses were performed using the Pandas[56,57], GeoPandas[58], and statmodels[59] Python modules.

The counties with a CSN or IMPROVE monitor represent a sample of the total number of US counties. Confidence intervals for the sample mean concentrations and $PM_{2.5}$ content, population-weighted means, and relative disparity were calculated through bootstrap resampling.

To test the geographic dependence of fine particulate metal concentrations, we grouped US states into the categories defined in Morello-Frosch and Jesdale[20] (plotted in Fig. S1). We used Moran's I[60] as a measure of global spatial autocorrelation and local Moran's I to indicate local clusters of elevated and low concentrations.

## Ethical Review

This research did not require ethical review as it utilized de-identified, publicly available data, which does not constitute human subjects research as defined at 45 CFR 46.102.

## Data availability

Observations of fine particulate metals from CSN and IMROVE are publicly available (http://views.cira.colostate.edu/fed/QueryWizard/Default.aspx). Demographic data at the county and census tract resolution are publicly available from the Census website (https://data.census.gov/cedsci/). The full dataset (in CSV format) required to reproduce the results described here has been deposited in the Mountain Scholar database under accession code: https://doi.org/10.25675/10217/235553.

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

## Acknowledgements

This work was supported financially by grants from the Health Effects Institute under grant number 4953- RFA14-3/16-4 awarded to FD, National Institute of Health under grant numbers DP2MD012722 and P50MD010428 awarded to FD, National Institute of Health and National Institute of Environmental Health Sciences under grant number R01 ES028033 awarded to FD, National Institute of Health and Columbia University under grant number 1R01ES030616 awarded to FD, the National Institute On Minority Health And Health Disparities of the National Institutes of Health under award number R01MD012769 awarded to MLB and FD, the Environmental Protection Agency under grant number 83587201-0 awarded to FD and grant number RD83587101 awarded to MLB, The Climate Change Solutions Fund, and the Harvard Star Friedman Award. This study has not been formally reviewed by EPA. The views expressed in this document are solely those of the authors and do not necessarily reflect those of the Agency. EPA does not endorse any products or commercial services mentioned in this publication. The content is solely the responsibility of the authors and does not necessarily represent the official views of the National Institutes of Health.

## Author contributions

J.K.K. and J.V. conceived of the study design and premise. J.K.K. integrated and analyzed data. J.K.K. wrote the manuscript with contributions from M.L.B., F.D., C.L., K.J.G.P., S.W., X.W., J.V. Finally, J.K.K., M.L.B., F.D., C.L., K.J.G.P., S.W., X.W., and J.V. contributed to research design, motivation, and scope.

## Competing interests

The authors declare no competing interests
