## [Peer Review File · Nature Communications]

Reviewer Comments, first round

Reviewer #1 (Remarks to the Author):

Key results

Racial residential segregation, quantified as the race-to-race dissimilarity index of tracts within a county, is associated with an increase in both anthropogenic and predominantly natural metal components of PM_{2.5}, as well as total PM_{2.5}. Relative disparity between high- and low-segregation counties is higher for anthropogenic metal PM_{2.5} components than for PM_{2.5} mass or other metals. Both RRS and metal PM exhibit geographic patterns determined by visual inspection, which reflect historical geographic trends in industrialization and population movement. A temporal analysis of vanadium concentrations shows that cities with higher RRS benefitted disproportionately from reductions due to recent policy.

Validity

The explicit results are valid, supported by high-quality data, and the limitations of both the data and analyses are adequately addressed. However, in some sense, the spirit of the article is not supported by the analyses as currently described. The mention of racial/ethnic exposure disparities and "injustices" in the abstract and introduction would suggest that the analysis would seek to investigate differences in exposure between non-Hispanic white, non-Hispanic Black, Hispanic, Asian, and other people of color (POC), as well as examine the interaction between RRS (as a signifier of city-scale racial discrimination/injustice) and racial/ethnic disparity, addressing the concept of environmental justice (EJ). The discussion also hints at the application of this data to racial/ethnic health disparities, noting that that higher metals concentrations "may contribute to higher risk or [sic] morbidity and mortality in segregated communities with high non-Hispanic Black populations" (236-237). However, the analysis does not include any quantification of racial/ethnic exposure disparity, with the relative disparity metrics discussed in text and displayed in Figures 3 and 4 apparently only considering differences within the total population, across RRS levels. It seems that the data as described and provided to reviewers would support the calculation of RD by race/ethnicity and the stratification of modeling results by race/ethnicity. This reviewer believes that the conclusions of the paper would be much stronger – and of interest to a broader audience – if such results were added to the paper to the extent that they would not unreasonably expand the current scope.

To address the idea that identifying an association between RRS and concentrations is in and of itself an EJ finding: It seems likely from established literature on PM_{2.5} and air toxics that the environmental racism embedded in US residential segregation – including but not limited to the tendency to restrict housing options for POC to areas nearer to industry and major transport corridors, and the establishment of zoning policies that place emissions-generating infrastructure in areas where majority-POC populations live, owing to their limited political power to fight such zoning – would mean a connection between RRS and higher exposure for POC. Based on patterns observed for other pollutants, within a highly segregated county, POC would likely experience higher anthropogenic PM_{2.5} metal exposure than white residents of the same county. However, this is not an assumption that may be taken for granted. In fact, it is an assumption that is somewhat undermined by the methodological decision to take the average of all monitors within a county, thus implying that exposure is uniform within a county. This reviewer acknowledges that drawing this conclusion would require additional data, either more measurement sites or a modeling method (e.g., LUR) to represent within-county variability, and is not within the scope of this paper. But, without proof of that conclusion, the findings of this paper as written carry much less weight as an example of environmental injustice. This reviewer does caution that using phrasing like "segregation-based disparities in vanadium exposure" (20-21, 284-285) may be interpreted as "racial/ethnic disparities in vanadium exposure within a city resulting from the disproportionate concentration of emissions activities and POC in the same neighborhoods," which conclusions are not supported by the analysis.

One final comment on the validity of the analysis is that while the authors mention both regional and urban/rural spatial patterns of metals emission sources (lines 159-160, 180), only the

urban/rural distinction is explored and not the broader pattern of spatial autocorrelation. This limits the interpretability of the model results but could be addressed with some adjustments in analysis methods.

Significance

This paper could present a significant advance in the field of EJ literature. While, as the authors note, there is a rich literature quantifying racial/ethnic disparity in exposure to criteria air pollutants (especially NO₂ and PM_{2.5}) and air toxics, and more limited literature investigating the additional effect of segregation on these disparities, this reviewer is not aware of any existing study quantifying disparity in exposure to the metal components of PM_{2.5}. As the authors compellingly emphasize, PM composition and metal content may represent higher particle toxicity than total PM mass and so the quantification of disparity in these components carries particular relevance for health disparities beyond what has already been established. However, as stated above, this reviewer believes the analysis in its current form stops short of a major EJ milestone, although it presents some findings on PM_{2.5} metal exposure that would be of interest to a more specialized audience.

Data and methodology

PM_{2.5} data: the CSN network provides high-quality measurements and the QA/QC procedures described in this paper are sound, including the choice to keep valid measurements below MDL. Although the authors note on lines 332-333 that the 223 counties including a CSN/IMPROVE monitor do not necessarily represent an unbiased/representative sample of US, this network still provides wide geographic coverage of both rural and urban conditions and is a very valuable source of PM composition data not available elsewhere.

A minor question for the authors is why Pb was chosen to represent anthropogenic metals when it had the greatest share (78%) of measurements were below MDL (Table S1)?

The choice of ACS population data is appropriate and well justified.

The dissimilarity index is a well-established metric for expressing residential segregation, although some comparable analyses (Morello-Frosch and Jesdale, 2005) have opted for an integrative metric for segregation across all race/ethnicities and this may simplify the presentation of results. This reviewer is less familiar with the use of CoV/RD as a metric for disparity compared with the normalized difference in population-weighted means, but its use is preceded in very recent literature.

Analytical approach

Both the results section providing a description of spatial patterns and the modeling of RRS-metals relationships would benefit from an explicit/quantitative treatment of spatial patterning. Although it is not strictly necessary, the descriptions of regional hot spots in lead concentrations would benefit from some more quantitative measures of spatial autocorrelation such as either LISA via local Moran's I or a Getis-Ord clustering analysis.

Accommodating spatial autocorrelation with the choice of statistical model is more important. Both the metal concentrations and DI show geographic patterns attributable to complex social/economic/political causes that may confound conclusions drawn from a simple linear model. Morello-Frosch and Jesdale (2005) noted that "because previous research has documented regional variation in both the level of racial/ethnic segregation and its causes (Frey and Farley 1996)", it was appropriate to control for regional factors by including it as an additional variable in a multivariate regression. Another option is to choose a model that accounts for the spatial structure of model error by including spatial lag in the error term of the model equation; although I do not analyze my own data using Python I know such utilities are readily available.

In addition to explicitly addressing spatial relationships in the analytical approach, this reviewer believes it would benefit the paper as a whole to consider both race/ethnicity and RRS as complementary factors potentially influencing metals concentrations. It appears that the current data would support using race (e.g., % NHB in a county) as a predictor in the model. Alternatively, models may be stratified by race ethnicity – Morello-Frosch and Jesdale (2005) demonstrate a

method of doing so with the ecologic unit of e.g., census tract rather than individual-level exposure. The authors may be aware of data limitations or pitfalls in such an approach, but the dual exploration of race/ethnicity and RRS is usefully explored in many of the works cited in this paper's introduction (Woo et al. 2019, Kravitz-Wirtz 2016, Jones et al. 2014, and aforementioned Morello-Frosch and Jesdale 2005), many of those studies also did not include complete coverage of the US, and none have yet considered PM2.5 composition.

Minor issues:

Are the pollution data used in the modeling log-transformed? The text in the caption of Table 2 indicating that the coefficient represents a percent increase in concentration corresponding to 10% increase in dissimilarity index seems to imply that they were, but I did not see mention of that in the methods text.

The authors note that "the spatial heterogeneity of pollution levels could further differ by source and chemical structure of particles (334-335)", and it would be very interesting to know what, if anything, CSN/IMPROVE network data reveals about this. In the cases where multiple monitors were located within the same county and averaged, it would be useful to describe in the SI the differences in measurements between monitors. Presumably, there are few multi-monitor counties, but within-county variability is highly relevant in the context of segregation potentially amplifying within-county racial/ethnic exposure disparity.

Suggested improvements

These are embedded in the "validity" and "analytical approach" sections above. This reviewer acknowledges that these verge on significantly changing the scope of this work and would require substantial additions to the text of the results and discussion section, but also believes that these additions would add tremendous value to the work.

Minor:

Equations are not numbered; I believe y in eq. 3 should be capitalized.

Figure S3, the lower whiskers on the box plot for "Well Integrated" counties in both the Pb and Fe panel seem to be different than those in Figure 2, although I believe they are representing the same data.

Clarity and context

The paper is well structured, well written, and results are described clearly.

References

The references provided are appropriate and demonstrate the authors' collective expertise in the fields of environmental justice, PM2.5 composition and toxicity, and PM exposure.

Reviewer #2 (Remarks to the Author):

This study examines the relationships between racial residential segregation and the specific toxic components of PM2.5. While most previous studies focus on the total PM2.5 concentrations, this study demonstrates how the different fine particulate metals, which are potentially toxic components in PM2.5, have different geographic distributions across the US depending on whether they are emitted from anthropogenic or natural sources. It then investigates how the fine particulate metal concentrations are associated with racial residential segregation. Overall, this paper reads well, and its findings are interesting. Below are my suggestions and questions.

1. The authors linked the county-level Dissimilarity index values to the point-level measurement data. However, the measurement at one specific monitor location in a county cannot represent the overall concentration for the entire county. PM2.5 concentrations vary significantly over space. Their spatial variation is typically higher than that of other pollutants including ground-level ozone. Furthermore, because the authors rely on this discrete point data, their analysis is limited to

counties with a CSN or IMPROVE monitor. Therefore, only a total of 233 counties, out of more than 3,000 counties in the US, were included in the analysis. Although I understand that it is difficult to obtain data with a high spatial resolution due to the limited number of monitors across the US, the authors could consider modeling continuous, smooth surfaces for Pb and Fe concentrations using the measurement data and spatial interpolation methods, such as IDW or Kriging. With the continuous surface raster data, they can better display the geographic distributions of different types of fine particulate metals than with the discrete point data (Fig 1). This surface raster data would also allow them to include many more counties in their statistical analysis, including some "unmonitored" rural and low-DI counties. For statistical analysis, they can calculate a county average using the values of all pixels in the county. Although it still may not fully capture local variations in the concentrations, I think that this county-level average may be better representative of concentrations of the county than a value measured at a single location in the county. I wonder if using county-level concentration data and adding more counties to their analysis would change their findings and conclusions.

2. The Dissimilarity index has several limitations. It has long been criticized as a "non-spatial" measure that does not capture spatial relationships between population groups and spatial units. This causes the checkerboard landscape problem, modifiable areal unit problem (MAUP), and uncertain geographic context problem (UGCoP) (Park & Kwan, 2017)*. These issues should be discussed as the limitations of this study.

*Park, Y.M. & Kwan, M.-P. (2017). Multi-Contextual Segregation and Environmental Justice Research: Toward Fine-Scale Spatiotemporal Approaches. *International Journal of Environmental Research and Public Health*, 14(10), 1205.

3. The authors categorized monitoring sites into two groups: urban vs rural sites. What are the criteria for this grouping? Please provide more details about how this was determined because this information is important to be fully convinced that their findings are not solely due to urban/rural differences in racial segregation.

4. Considering their interesting findings, their title sounds a bit too generic to me. I believe their major finding is that the strength of the association with racial residential segregation differs by different toxic components of PM_{2.5}. I would suggest slightly modifying the title to better highlight their noteworthy findings. (in addition, although it's particulate metals, not particulate matter, I unconsciously read it as particulate matter because I was more familiar with particulate matter than particulate metals, and I did not recognize it until I began to read the body.)

We thank both reviewers for their helpful comments and suggested references. We have addressed all comments in a revised manuscript, and we provide a point-by-point response in this document. Text for the reviewers are shown in italics, our response in plain text, and revisions are shown in bold.

Response to Reviewer 1

1. The explicit results are valid, supported by high-quality data, and the limitations of both the data and analyses are adequately addressed. However, in some sense, the spirit of the article is not supported by the analyses as currently described. The mention of racial/ethnic exposure disparities and "injustices" in the abstract and introduction would suggest that the analysis would seek to investigate differences in exposure between non-Hispanic white, non-Hispanic Black, Hispanic, Asian, and other people of color (POC), as well as examine the interaction between RRS (as a signifier of city-scale racial discrimination/injustice) and racial/ethnic disparity, addressing the concept of environmental justice (EJ). The discussion also hints at the application of this data to racial/ethnic health disparities, noting that that higher metals concentrations "may contribute to higher risk or [sic] morbidity and mortality in segregated communities with high non-Hispanic Black populations" (236-237). However, the analysis does not include any quantification of racial/ethnic exposure disparity, with the relative disparity metrics discussed in text and displayed in Figures 3 and 4 apparently only considering differences within the total population, across RRS levels. It seems that the data as described and provided to reviewers would support the calculation of RD by race/ethnicity and the stratification of modeling results by race/ethnicity. This reviewer believes that the conclusions of the paper would be much stronger – and of interest to a broader audience – if such results were added to the paper to the extent that they would not unreasonably expand the current scope.

We thank the reviewer for this constructive suggestion and agree that exploring the possibility of racial and ethnic disparities in exposure (and the interaction with racial residential segregation) would strengthen the main conclusions of our study. To this end, we added several analyses that explore this concept. First, we added a figure in the Supplemental Material (Figure S5) showing the distributions of PM_{2.5} and fine particulate metals in counties with a non-Hispanic White, non-Hispanic Black, Asian, Native American, or Hispanic population fraction higher than the respective national average. This provides evidence that counties with above-average non-Hispanic White population fraction are exposed to lower concentrations of PM_{2.5} and fine particulate metals than counties with an above average POC population fraction. It also allows for a direct comparison with previous studies that rely on population fraction of racial and ethnic groups (for instance, this analysis follows that described in Jbailey et al., 2022). Next, to explore the possible interaction between RRS and racial/ethnic disparities, we expanded our regression analysis to include a multivariable regression of fine particulate metal concentrations with RRS and proportion of racial/ethnic population (to treat these as confounding factors). We found a statistically significant positive association between fine particulate metal concentrations with RRS and proportion of residents identifying as non-Hispanic Black (Tables S5 and S6). Thus, across counties with similar levels of RRS, fine particulate metal concentrations tend to increase with increasing percent of non-Hispanic Black residents. We have added the following statements to the main text:

In addition to this association of the DI with fine particulate metals, we also find an association between race/ethnicity and fine particulate metal concentrations. In counties where the percent of the population that identifies as NHB is greater than the national average across all counties, the concentrations of fine particulate metals are consistently elevated relative to counties with a higher

than average NHW or Native American population (Figure S6). This same association is also seen for PM_{2.5}, in good agreement with previous studies⁴⁸. Further, we examine the interaction between RRS and racial/ethnic group makeup in each county by expanding the linear model to include an additional variable for the percent of the population identifying as NHB (and a separate model for NHW). We find that concentrations of fine particulate metals commonly associated with anthropogenic emissions increase by 4-8% per 10% increase in DI and 4-6% per 10% increase in NHB population (Table S5). Conversely, these same metals increase by 9-15% per 10% increase in DI yet decrease by 4-10% per 10% increase in NHW population (though the latter coefficient is not statistically significant for all metals; Table S6). Thus, counties with a high degree of RRS and high NHB population tend to be exposed to higher concentrations of fine particulate metals than counties with a high degree of RRS and a high NHW population.

2. To address the idea that identifying an association between RRS and concentrations is in and of itself an EJ finding: It seems likely from established literature on PM_{2.5} and air toxics that the environmental racism embedded in US residential segregation – including but not limited to the tendency to restrict housing options for POC to areas nearer to industry and major transport corridors, and the establishment of zoning policies that place emissions-generating infrastructure in areas where majority-POC populations live, owing to their limited political power to fight such zoning – would mean a connection between RRS and higher exposure for POC. Based on patterns observed for other pollutants, within a highly segregated county, POC would likely experience higher anthropogenic PM_{2.5} metal exposure than white residents of the same county. However, this is not an assumption that may be taken for granted. In fact, it is an assumption that is somewhat undermined by the methodological decision to take the average of all monitors within a county, thus implying that exposure is uniform within a county. This reviewer acknowledges that drawing this conclusion would require additional data, either more measurement sites or a modeling method (e.g., LUR) to represent within-county variability, and is not within the scope of this paper. But, without proof of that conclusion, the findings of this paper as written carry much less weight as an example of environmental injustice. This reviewer does caution that using phrasing like "segregation-based disparities in vanadium exposure" (20-21, 284-285) may be interpreted as "racial/ethnic disparities in vanadium exposure within a city resulting from the disproportionate concentration of emissions activities and POC in the same neighborhoods," which conclusions are not supported by the analysis.

The lack of within-county estimates of the variability of fine particulate metal concentrations is a limitation of this analysis. Unfortunately, the EPA monitoring network does not have the spatial resolution to provide an estimate of disparities within counties. Of the EPA monitoring sites which meet our inclusion criteria, only 22 counties have more than one monitor (and only four of these have more than 2 monitors). As the reviewer acknowledges, using additional modeling methods to represent within-county variability is beyond the scope of this paper, especially given the difficulties due to the sparsity of the sampling network. However, we do feel that the additional analysis included above (finding that segregated counties with a higher percent of the population identifying as non-Hispanic Black tend to have higher concentrations of fine particulate metals) strengthens our argument. We have added the following statement to the Methods section highlighting the relatively few number of counties with more than one CSN/IMPROVE monitor:

In counties with more than one CSN or IMPROVE monitor, concentrations of the respective pollutants were averaged across the monitors. In 2019, only 21 counties had more than one CSN or IMPROVE monitor (which met our inclusion criteria) and of these only six had more than two monitors. The relative standard deviation in concentration in fine particulate metals and total PM_{2.5} mass ranged from 0.3-140% across these counties with more than one monitor indicating that there is likely important variability in fine particulate metal concentrations within each county.

In addition, we have added the following to the discussion of limitations of this study:

Importantly, the spatial coverage often does not include more than one monitoring site within a county making quantifying within-county variability challenging...Expanding the CSN/IMPORVE network to allow for estimates of within-county variability along with additional rural and low-DI counties would strengthen subsequent studies.

We appreciate the reviewer pointing out possible interpretations of the phrase “segregation-based disparities”. We have revised these sentences to read:

For example, recent regulations on marine fuel oil not only reduced vanadium concentrations in coastal cities, but also sharply lessened differences in vanadium exposure by segregation.

3. One final comment on the validity of the analysis is that while the authors mention both regional and urban/rural spatial patterns of metals emission sources (lines 159-160, 180), only the urban/rural distinction is explored and not the broader pattern of spatial autocorrelation. This limits the interpretability of the model results but could be addressed with some adjustments in analysis methods.

We agree that further analysis of regional variability (including spatial autocorrelation) would strengthen our modeling results. We have expanded our modeling and analysis to include regional spatial patterns of concentration. We discuss our additions in response to comments 9 and 10.

Significance

4. This paper could present a significant advance in the field of EJ literature. While, as the authors note, there is a rich literature quantifying racial/ethnic disparity in exposure to criteria air pollutants (especially NO₂ and PM_{2.5}) and air toxics, and more limited literature investigating the additional effect of segregation on these disparities, this reviewer is not aware of any existing study quantifying disparity in exposure to the metal components of PM_{2.5}. As the authors compellingly emphasize, PM composition and metal content may represent higher particle toxicity than total PM mass and so the quantification of disparity in these components carries particular relevance for health disparities beyond what has already been established. However, as stated above, this reviewer believes the analysis in its current form stops short of a major EJ milestone, although it presents some findings on PM_{2.5} metal exposure that would be of interest to a more specialized audience.

We agree, please see our response to comment 1.

Data and methodology

5. PM_{2.5} data: the CSN network provides high-quality measurements and the QA/QC procedures described in this paper are sound, including the choice to keep valid measurements below MDL. Although

the authors note on lines 332-333 that the 223 counties including a CSN/IMPROVE monitor do not necessarily represent an unbiased/representative sample of US, this network still provides wide geographic coverage of both rural and urban conditions and is a very valuable source of PM composition data not available elsewhere.

6. A minor question for the authors is why Pb was chosen to represent anthropogenic metals when it had the greatest share (78%) of measurements were below MDL (Table S1)?

We chose Pb for its well known health risks and prevalence in scientific literature and new articles. We find that our overall conclusions and discussion do not change if we were to use a different metal to represent anthropogenic metals (such as Zn).

7. The choice of ACS population data is appropriate and well justified.

8. The dissimilarity index is a well-established metric for expressing residential segregation, although some comparable analyses (Morello-Frosch and Jesdale, 2005) have opted for an integrative metric for segregation across all race/ethnicities and this may simplify the presentation of results. This reviewer is less familiar with the use of CoV/RD as a metric for disparity compared with the normalized difference in population-weighted means, but its use is preceded in very recent literature.

We thank the reviewer for the suggestion of an alternate metric for segregation across combined racial and ethnic groups. We calculated this metric following the discussion in Morello-Frosch and Jesdale (2006) and tested this metric in our modeling approach. We find a similar statistically significant positive correlation of fine particulate metal concentrations with RRS. While we have chosen to continue to use our original dissimilarity index for racial/ethnic groups individually, we have included the following statement showing that our conclusions are similar using the integrative metric (where the superscript is a reference to Morello-Frosch and Jesdale, 2006):

In addition, we tested an integrated dissimilarity index intended to capture the degree of RRS across all racial and ethnic groups²⁰; this approach yielded similar conclusions.

Analytical approach

9. Both the results section providing a description of spatial patterns and the modeling of RRS-metals relationships would benefit from an explicit/quantitative treatment of spatial patterning. Although it is not strictly necessary, the descriptions of regional hot spots in lead concentrations would benefit from some more quantitative measures of spatial autocorrelation such as either LISA via local Moran's I or a Getis-Ord clustering analysis.

We thank the reviewer for their constructive comments regarding quantitative spatial analysis. We have included a number of additional analyses. First, we separated our measurements into the geographic regional groupings of states suggested in Morello-Frosch and Jesdale (2006). We have added a figure to the Supplemental Material (Figures S1 and S2) examining the distribution of concentrations and PM_{2.5} mass proportions of fine particulate metals across these regions. We also calculated measures of global and local spatial autocorrelation (Table S1 and Figure S3). This analysis supports a statistically significant spatial structure to lead concentrations (i.e., a significant global spatial autocorrelation) as well as regional

hot spots (notably in the Midwest/Northeast) and cold spots (in the Mountain and Border states). Iron appears to have a lower degree of spatial structure as evidenced by the lower Moran's I measure of global spatial autocorrelation and the diminished clustering compared to lead. We have revised the following discussion to include this analysis:

Concentrations of fine particulate Pb show a strong geographical distribution, with a statistically significant degree of global spatial autocorrelation (Figure 1 and Table S1). We find a statistically significant cluster of elevated concentration in the industrial Midwest near the Ohio River Valley, with an average concentration of 3 ng m⁻³, and low concentrations in the Western, Mountain, and Border states, with an average concentration of 1 ng m⁻³ (Figure 1 and Figures S2-3).

In contrast, fine particulate Fe concentrations display a lower degree of spatial dependence (Table S1 and Figure S2-3). Median Fe concentrations in Border states (51 ng m⁻³) are slightly lower than concentrations in the Midwest (78 ng m⁻³). However, after normalizing to PM_{2.5} mass, the mass proportion of Fe is highest in the Border states, likely reflecting the mineral dust source of Fe in the desert Southwest (Figure 1 and Figure S2).

10. Accommodating spatial autocorrelation with the choice of statistical model is more important. Both the metal concentrations and DI show geographic patterns attributable to complex social/economic/political causes that may confound conclusions drawn from a simple linear model. Morello-Frosch and Jesdale (2005) noted that "because previous research has documented regional variation in both the level of racial/ethnic segregation and its causes (Frey and Farley 1996)", it was appropriate to control for regional factors by including it as an additional variable in a multivariate regression. Another option is to choose a model that accounts for the spatial structure of model error by including spatial lag in the error term of the model equation; although I do not analyze my own data using Python I know such utilities are readily available.

Thank you for this suggestion. We tested both models (one which includes a categorical variable for the regional state groupings defined in Morello-Frosch and Jesdale and a second which accounts for spatial lag in the error term). In both cases, we still find a statistically significant positive association of DI with fine particulate metal concentrations. We now reference the model that includes a fixed effect categorical variable (Table S3) for the regional grouping of states, as this is consistent with the discussion in paragraph as a whole:

We find a similar relationship after controlling for the geographic regions discussed above (Table S3).

11. In addition to explicitly addressing spatial relationships in the analytical approach, this reviewer believes it would benefit the paper as a whole to consider both race/ethnicity and RRS as complementary factors potentially influencing metals concentrations. It appears that the current data would support using race (e.g., % NHB in a county) as a predictor in the model. Alternatively, models may be stratified by race ethnicity – Morello-Frosch and Jesdale (2005) demonstrate a method of doing so with the ecologic unit of e.g., census tract rather than individual-level exposure. The authors may be aware of data limitations or pitfalls in such an approach, but the dual exploration of race/ethnicity and RRS is usefully explored in many of the works cited in this paper's introduction (Woo et al. 2019, Kravitz-Wirtz

2016, Jones et al. 2014, and aforementioned Morello-Frosch and Jesdale 2005), many of those studies also did not include complete coverage of the US, and none have yet considered PM_{2.5} composition.

Thank you for this comment. We have included this analysis in our study (our specific additions were outlined in our response to comment 1).

Minor issues:

12. *Are the pollution data used in the modeling log-transformed? The text in the caption of Table 2 indicating that the coefficient represents a percent increase in concentration corresponding to 10% increase in dissimilarity index seems to imply that they were, but I did not see mention of that in the methods text.*

Yes, the concentration of the pollutants are log-transformed for modeling. We have added the following statement to the text to clarify this:

We used a univariate linear regression model to test the strength of the association between DI and particulate metal (and total PM_{2.5} mass) concentrations (where both DI and pollutant concentrations have been log-transformed).

13. *The authors note that "the spatial heterogeneity of pollution levels could further differ by source and chemical structure of particles (334-335)", and it would be very interesting to know what, if anything, CSN/IMPROVE network data reveals about this. In the cases where multiple monitors were located within the same county and averaged, it would be useful to describe in the SI the differences in measurements between monitors. Presumably, there are few multi-monitor counties, but within-county variability is highly relevant in the context of segregation potentially amplifying within-county racial/ethnic exposure disparity.*

Of the counties with a CSN/IMPROVE monitor considered in this study (in the year 2019), only 21 had more than one monitor and only six of these had more than 2 monitors within the county. To get a sense of the within-county variability measured by the monitors, we calculated the relative standard deviation (expressed as a percent) for total PM_{2.5} mass and the fine particulate metals; however, we note that as most of these counties contain only two monitors, this analysis is limited. We find that for PM_{2.5} mass concentrations the relative standard deviation across counties with more than one monitor ranges from 0.4% to 84%. The relative standard deviation in fine particulate metal concentrations across counties with more than one monitor ranged from 0.3% to 143%. Higher spatial resolution measurement networks (or model estimates) of fine particulate metal concentrations would certainly be a useful area of future work. We have added a table of the relative standard deviations for each pollutant for each county in the Supplemental Material and have added the following statement to the main text:

In counties with more than one CSN or IMPROVE monitor, concentrations of the respective pollutants were averaged across the monitors. In 2019, only 21 counties had more than one CSN or IMPROVE monitor (which met our inclusion criteria) and of these only six had more than two monitors. The relative standard deviation in concentration in fine particulate metals and total PM_{2.5} mass ranged from 0.3-140% across these counties with more than one monitor indicating that there is likely important variability in fine particulate metal concentrations within each county.

In addition, we have added the following to the discussion of limitations of this study:

Importantly, the spatial coverage often does not include more than one monitoring site within a county making quantifying within-county variability challenging...Expanding the CSN/IMPORVE network to allow for estimates of within-county variability along with additional rural and low-DI counties would strengthen subsequent studies.

Suggested improvements

14. *These are embedded in the "validity" and "analytical approach" sections above. This reviewer acknowledges that these verge on significantly changing the scope of this work and would require substantial additions to the text of the results and discussion section, but also believes that these additions would add tremendous value to the work.*

We agree and are happy with the resulting additions to our study.

Minor:

15. *Equations are not numbered; I believe y in eq. 3 should be capitalized.*

We have made these changes.

16. *Figure S3, the lower whiskers on the box plot for "Well Integrated" counties in both the Pb and Fe panel seem to be different than those in Figure 2, although I believe they are representing the same data.*

We apologize for this error. These are representing the same data. We have corrected this figure.

Clarity and context

17. *The paper is well structured, well written, and results are described clearly.*

References

18. *The references provided are appropriate and demonstrate the authors' collective expertise in the fields of environmental justice, PM_{2.5} composition and toxicity, and PM exposure.*

Response to Reviewer 2

This study examines the relationships between racial residential segregation and the specific toxic components of PM_{2.5}. While most previous studies focus on the total PM_{2.5} concentrations, this study demonstrates how the different fine particulate metals, which are potentially toxic components in PM_{2.5}, have different geographic distributions across the US depending on whether they are emitted from anthropogenic or natural sources. It then investigates how the fine particulate metal concentrations are associated with racial residential segregation. Overall, this paper reads well, and its findings are interesting. Below are my suggestions and questions.

1. *The authors linked the county-level Dissimilarity index values to the point-level measurement data. However, the measurement at one specific monitor location in a county cannot represent the overall concentration for the entire county. PM_{2.5} concentrations vary significantly over space. Their spatial variation is typically higher than that of other pollutants including ground-level ozone. Furthermore,*

because the authors rely on this discrete point data, their analysis is limited to counties with a CSN or IMPROVE monitor. Therefore, only a total of 233 counties, out of more than 3,000 counties in the US, were included in the analysis. Although I understand that it is difficult to obtain data with a high spatial resolution due to the limited number of monitors across the US, the authors could consider modeling continuous, smooth surfaces for Pb and Fe concentrations using the measurement data and spatial interpolation methods, such as IDW or Kriging. With the continuous surface raster data, they can better display the geographic distributions of different types of fine particulate metals than with the discrete point data (Fig 1). This surface raster data would also allow them to include many more counties in their statistical analysis, including some “unmonitored” rural and low-DI counties. For statistical analysis, they can calculate a county average using the values of all pixels in the county. Although it still may not fully capture local variations in the concentrations, I think that this county-level average may be better representative of concentrations of the county than a value measured at a single location in the county. I wonder if using county-level concentration data and adding more counties to their analysis would change their findings and conclusions.

We agree with the reviewer that having more measurement sites within counties (increasing within-county variability) and more counties with measurement sites (increasing the number of counties included in the analysis) would aid our study; however, the sampler density of the CSN and IMPROVE networks is a limitation of this study. In 2019, there were only 21 counties with more than one CSN/IMPROVE monitor (which met our inclusion criteria), and of these only 6 had more than 2 monitors. In these counties, the relative standard deviation of fine particulate metal concentrations range from 0.3% to 143%. While this shows there can be variability of metal concentrations within counties, we currently lack the measurement network to capture this variability across all counties in our study. Moreover, the lack of measurement sites makes attempting a spatial interpolation (such as kriging) inappropriate.

We agree that our study includes a fraction of the total number of counties across the US and that this is an inherent limitation in our analysis. However, we note that using modeled data would introduce its own important limitations and pitfalls due to the lack of long-term measurement sites with which to validate. Instead, we address this limitation in the following ways. First, we acknowledge this limitation and compare the demographics of the measurement sites included in the study to the national average (lines 259-263 in the original submission). This provides an interesting motivation for expanding the EPA measurement networks. We have added statements to the main text to expand on these results. Second, we estimate uncertainty ranges on all calculations using bootstrapping (lines 351-353). This approach is intended to estimate uncertainty in the true mean given repeated sampling of a smaller population. While this is not perfect, the uncertainty range provides some estimate on how changing the sample of counties may change the overall analysis results.

We have added the following:

Importantly, the spatial coverage often does not include more than one monitoring site within a county making quantifying within-county variability challenging...Expanding the CSN/IMPORVE network to allow for estimates of within-county variability along with additional rural and low-DI counties would strengthen subsequent studies.

2. *The Dissimilarity index has several limitations. It has long been criticized as a “non-spatial” measure that does not capture spatial relationships between population groups and spatial units. This causes the checkerboard landscape problem, modifiable areal unit problem (MAUP), and uncertain geographic context problem (UGCoP) (Park & Kwan, 2017)*. These issues should be discussed as the limitations of this study.*

**Park, Y.M. & Kwan, M.-P. (2017). Multi-Contextual Segregation and Environmental Justice Research: Toward Fine-Scale Spatiotemporal Approaches. International Journal of Environmental Research and Public Health, 14(10), 1205.*

This is an excellent paper and we thank the reviewer for this recommendation. We have added the following statement discussing the limitations of the dissimilarity index (where the superscript references Park and Kwan, 2017):

Finally, in calculating DI and degrees of RRS, we rely on arbitrary geographies, census tracts and counties, which may not accurately reflect the spatial distributions of neighborhoods. The DI has noted limitations as an aspatial index of segregation in that the index for a given county is invariant to the spatial distribution of the census tracks and values of the DI may be different with a different choice of spatial scale⁵⁵. Further, this metric relies solely on place of residence and does not take into account the time people may spend in different neighborhoods (such as for work).

3. *The authors categorized monitoring sites into two groups: urban vs rural sites. What are the criteria for this grouping? Please provide more details about how this was determined because this information is important to be fully convinced that their findings are not solely due to urban/rural differences in racial segregation.*

We categorized urban and rural sites based on an urban areas geographic dataset publicly available from the US Census Bureau. We have added the following to explicitly state this in the main text:

Monitoring sites were classified as urban or rural based on urban area classifications from the US Census Bureau.

4. *Considering their interesting findings, their title sounds a bit too generic to me. I believe their major finding is that the strength of the association with racial residential segregation differs by different toxic components of PM2.5. I would suggest slightly modifying the title to better highlight their noteworthy findings. (in addition, although it's particulate metals, not particulate matter, I unconsciously read it as particulate matter because I was more familiar with particulate matter than particulate metals, and I did not recognize it until I began to read the body.)*

Thank you for this suggestion. We appreciate the importance of highlighting our results in our title. We have revised the title to the following which we feel better captures our most important result:

Unequal airborne exposure burden to toxic metals is associated with race, ethnicity, and segregation

Reviewer Comments, second round

Reviewer #1 (Remarks to the Author):

The authors have adequately addressed my previous concerns. I appreciate the consideration and additional effort taken to make those improvements.

Reviewer #2 (Remarks to the Author):

I feel that the authors have done a creditable job in responding to all issues that I and the other reviewer raised and that the current version is acceptable for publication. It will inspire the Nature Communications readership.

We thank both reviewers for their time and consideration of this study.

Reviewer #1:

The authors have adequately addressed my previous concerns. I appreciate the consideration and additional effort taken to make those improvements.

Thank you for your previous suggestions. We feel these comments improved the paper.

Reviewer #2:

I feel that the authors have done a creditable job in responding to all issues that I and the other reviewer raised and that the current version is acceptable for publication. It will inspire the Nature Communications readership.

Thank you for your comments and careful consideration of this study. We feel the points brought up in revisions strengthened our study.